

# Near sea ice-free conditions in the northern route of the Northwest Passage at the end of the 2024 melt season

Stephen E.L. Howell[1], Alex Cabaj[1], David G. Babb[2], Jack C. Landy[3], Jackie Dawson[4], Mallik Mahmud[5], and Mike. Brady[1]

[1]Climate Research Division, Environment and Climate Change Canada, Toronto, ON, Canada
[2]Centre for Earth Observation Science, University of Manitoba, Winnipeg, MB, Canada
[3]Department of Physics and Technology, The Arctic University of Norway, Tromsø, Norway
[4]Department of Geography, University of Ottawa, Ottawa, ON, Canada
[5]Department of Geography, McGill University, Montreal, QC, Canada

*Correspondence to*: S.E.L. Howell (stephen.howell@ec.gc.ca)

**Abstract.** The Northwest Passage through the Canadian Arctic Archipelago (CAA) provides a shorter transit route connecting the Atlantic Ocean to the Pacific Ocean but ever-present sea ice has prevented its practical navigation. Sea ice area in the northern route of the Northwest Passage on September 30, 2024 fell to a minimum of $4 \times 10^3$ km$^2$, the lowest ice area observed since 1960. Here, we investigate the processes responsible for the record low sea ice area in 2024 and show it was driven by a perfect sequence of thermodynamic and dynamic forcing events acting on an increasingly less resilient ice cover. Specifically, multi-year ice (MYI) only made up ~10% of total sea ice area at the start of the melt season that was characterized by an atmospheric circulation pattern that brought warm southerly air directly into the middle of CAA. This resulted in a record summer air temperature anomaly of 2.1℃ that drove rapid melt and limited the import of ice from higher latitude regions to 50% of the 2016-2024 mean. Finally, positive air temperature anomalies upwards of 12℃ persisted into October, extending the melt season further and delaying freeze-up by 1-month, compared to the 1991-2020 baseline. Overall, a series of specific and cascading thermodynamic and dynamic processes is required to melt all ice in the northern route of Northwest Passage as it did in 2024, therefore ice conditions along this route will likely continue to remain highly variable during the transition to a summertime sea ice free Arctic.

## 1 Introduction

The Northwest Passage through the Canadian Arctic Archipelago (CAA) is a maritime trade route connecting the Atlantic and Pacific Ocean. This long desired trade route is shorter and potentially more economical than current maritime routes through the Panama Canal, Suez Canal, and around the island of Cape Horn. Ever-present sea ice has thus far prevented practical utilization of the NWP, which includes two primary routes; a preferred shorter deepwater northern route directly through the Parry Channel from Baffin Bay to the Beaufort Sea and a shallow water southern route that runs south of Victoria Island (Figure 1). The trillion-dollar international shipping sector supports almost 90% of international trade globally is the 'greenest' form of transport from the perspective of greenhouse gases per goods moved. Thus, shipping is vital to modern society and is expected to continue to be a major driver of economic development. Recent nationalistic rhetoric, geopolitical tensions in the middle east, Africa, and Russia are likely to affect supply-chain resilience in the near-term future and could drive interest and



demand towards Arctic shipping routes including the Northwest Passage, especially as climate change continues influence
reductions in sea ice across the Arctic.

Sea ice extent and area across the Arctic have been decreasing over the past 40+ years (Parkinson and DiGirolamo, 2021). In
addition, the sea ice has become thinner and younger as thicker multi-year ice (MYI) has been replaced by thinner first-year
ice (FYI) (Comiso, 2012; Babb et al., 2023; Sumata et al., 2023; Babb and Howell, 2024; Krumpen et al., 2025). Future
projections by the latest state-of-the-art climate models indicate that at about mid-century a sea ice-free Arctic will be realized
(Notz & SIMIP Community, 2020; Jahn et al., 2024; Fol et al., 2025). As a result, many studies have suggested shipping
through the Northwest Passage will be more feasible (Smith and Stephenson, 2013; Mudryk et al., 2021) although increasingly
mobile sea ice may continue to present navigational safety challenges for operators (Howell and Haas 2015; Nicoll et al.,
2025). Indeed, shipping activity has increased within the Canadian Arctic as sea ice has thinned (Glissenaar et al., 2023) and
declined in area (Howell et al., 2023) but the northern route of the Northwest Passage is still typically avoided (Pizzolato et
al., 2016; Dawson et al., 2018; Nicoll et al., 2024). This is because there are regions along the northern route of the Northwest
Passage that contain high concentrations of MYI during the summer months and a direct path east to west is typically not
possible (Cook et al., 2024). These regions have been referred to as choke points and represent barriers to efficient or complete
transit of the northern route of the Northwest Passage. Moreover, ice floes that are thicker than 4 m and more than 100 m wide,
representing an impenetrable barrier to almost all ships, are still found within the northern route of the Northwest Passage
(Haas and Howell, 2015).

Since 2007, sea ice area in the northern route of the Northwest Passage has been lower when compared to the longer-term
observational record which begins in the 1960s (Tivy et al., 2011; Howell et al., 2023). Annual average air temperature in
northern Canadian regions has increased by 2.3°C over the period 1948–2016 (Zhang et al., 2019). However, warmer
temperatures driving melt are often insufficient to clear the northern route of the Northwest Passage because generally-thicker
sea ice is transported southward from higher latitudes and must also melt through completely (Howell et al., 2009; Howell et
al., 2013). As a result, there has been considerable interannual variability in ice conditions within the northern route of the
Northwest Passage, with exceptionally light ice observed in 1998 (Atkinson et al., 2006) and as far back as 1962 (Black, 1965;
Tivy et al., 2011). 2007 was perhaps the first time in human memory that the northern route of Northwest Passage was virtually
clear of ice (Howell et al., 2009). 2011 was even more dramatic with ice area considerably lower than 2007 marking the
previous record low (Howell et al., 2013). At the end of September in 2024, there was virtually no sea ice present in in the
northern route of the Northwest Passage ($4\times10^3$ km$^2$) marking the lowest ever observed area of since 1960 (Figure 1).

Here, we place the record low of 2024 in context within the longer-term record that extends back to 1960. We first discuss the
long-term spatial and temporal variability of minimum sea ice area years within the northern route of the Northwest Passage.





We then discuss the weekly evolution of sea ice area in 2024 as compared to previous low ice years. Finally, we discuss the factors that contributed to the record low of 2024.

**Figure 1:** Map of the routes of the Northwest Passage through the Canadian Arctic Archipelago. Also shown is the total sea ice concentration (%) on September 30, 2024.

## 2 Data and methods

### 2.1 Digital ice charts

The primary dataset used in this analysis is ice concentration and area from the Canadian Ice Service (CIS) digital ice charts and is available at and the data is available at: https://iceweb1.cis.ec.gc.ca/Archive/page1.xhtml?lang=en. For each available ice chart, we extracted the weekly time series of total and MYI area within the northern route of the Northwest Passage (see Figure 1) from 1968 to 2024. The CIS digital ice charts are a weekly compilation that integrates all available real-time sea ice information from various satellite sensors, aerial reconnaissance, ship reports, operational model results and the expertise of






experienced ice forecasters, since 1968 (Canadian Ice Service, 2007; Tivy et al., 2011). We also used digitized ice charts complied from the Polar Continental Shelf Project (PCSP) from 1960-1974. As with the CIS ice charts, aerial surveys together with all available remotely sensed and ground observation data were used in their production (Lindsay, 1976; Lindsay, 1977). The main concern with using digital ice charts to quantify long-term change is that the source information used in ice chart preparation also changes with time (Canadian Ice Service, 2007) but this bias is unquantifiable. As a result, concentration

estimates are potentially overestimated when satellite information was not routinely available. (i.e. the 1960s and 1970s). However, the majority of our analysis takes place in the RADARSAT era (i.e. 1995 to present) which greatly improves the quality of the ice information (Canadian Ice Service, 2007).

## 2.2 Ice area flux

We make use of the sea ice area flux record derived by Howell et al. (2024) and updated until September 2024 at the following

gates: M'Clure Strait, Fitzwilliam Strait, Byam Martin Channel, and Penny Strait. The M'Clure Strait was selected to quantify the ice area flux from the Beaufort Sea and the Fitzwilliam Strait, Byam Martin Channel, and Penny Strait to quantify the ice area flux from the Queen Elizabeth Islands (Figure 1). Briefly, synthetic aperture radar (SAR) imagery from RADARSAT-2, Sentintel-1, and the RADARSAT Constellation Mission (RCM) at HH polarization are ingested into the Environment and Climate Change Canada Automated Sea Ice Tracking System (ECCC-ASITS; Howell et al., 2022) that is based on the

Komarov and Barber (2013) feature tracking algorithm. The resulting sea ice motion output is sampled across each gate together with ice concentration from the CIS ice charts in order to determine the sea ice area flux across each gate. For all gates, the sea ice area flux values were summed over each month from October 2016 to September 2024. Positive values indicate import into the region and negative values indicate export out of the region. The monthly uncertainties are 4797 km$^2$ at the M'Clure Strait, 214 km$^2$ at Fitzwilliam Strait, 217 km$^2$ at Byam Martin Channel, and 257 km$^2$ at Penny Strait (Howell

et al., 2024).

## 2.3 Melt onset timing

We estimated the timing of melt onset in the northern route of the Northwest Passage using the methodology Mahmud et al. (2016) applied to Sentinel-1 synthetic aperture radar (SAR) imagery from 2016-2024. Briefly, the approach utilizes a pixel-based threshold method, capitalizing on the time series C-band SAR imagery backscatter values. This algorithm effectively

identifies melt onset timing on sea ice by analyzing backscatter variability exceeding specified threshold parameters compared to the mean winter backscatter. The melt onset timing is constrained using 2m air temperature to prevent false detection. We computed the spatial mean melt onset date within the northern route of the Northwest Passage from 2016-2024.

## 2.4 ERA reanalysis and APP-x

We also made use of monthly sea level pressure (SLP), 2-metre air temperature and total column vertically-integrated heat

flux        (eastward        and        northward)        from        ERA5        (Hersbach        et        al.,        2023)        available        at



https://cds.climate.copernicus.eu/datasets/reanalysis-era5-single-levels-monthly-means. Vertically integrated heat flux refers to the vertical integral (over pressure levels) of horizontal dry heat (enthalpy) fluxes, and has units of W/m. Standardized anomalies are calculated from reanalysis-derived quantities by dividing the climatological anomalies by the standard deviation of the 1991-2020 climatology.


Finally, we used albedo from the extended Advanced Very High Resolution Radiometer (AVHRR) Polar Pathfinder (APP-x; Wang & Key, 2005) from 1982 to 2024 available at: https://www.ncei.noaa.gov/access/metadata/landing-page/bin/iso?id=gov.noaa.ncdc:C00941 The data is available at a spatial resolution of 25 km and the uncertainty for albedo is 0.10 (Wang & Key, 2005).

**3. Results and Discussion**

**3.1 Long term spatial and temporal variability in minimum sea ice area**

The time series of the annual minimum sea ice area within the northern route of the Northwest Passage, from 1960 to 2024, is shown in Figure 2. In the 1960s-90s, the annual minimum ice area of ~$100 \times 10^3$ km$^2$ still covered ~60% of the total area of the northern route of the Northwest Passage. Interannual variability (standard deviation of $32 \times 10^3$ km$^2$) is considerable and ice

area ranged from a maximum ice area of $135 \times 10^3$ km$^2$ in 1979 to a minimum ice area of $4 \times 10^3$ km$^2$ in 2024. The second lowest ice area was in 2011 at $8 \times 10^3$ km$^2$ followed by 2023 at $10 \times 10^3$ km$^2$. Over the entire period from 1960 to 2024 a decreasing trend is apparent and over the 1968-2024 period from the CIS ice charts, the weekly sea ice area has decreased at a rate of 1379 km$^2$/year (Figure 2). Over the past 57 years the decline based on the CIS ice charts corresponds to a loss of $79 \times 10^3$ km$^2$, which is ~45% of the total area ($175 \times 10^3$ km$^2$) of the northern route of the Northwest Passage.


Interannual variability of sea ice area has been higher since 1998 (standard deviation of $33 \times 10^3$ km$^2$) when compared to the early part of the record (standard deviation of $22 \times 10^3$ km$^2$) with considerably larger sea ice area oscillations (Figure 2). Interestingly, up until 1998 there was a slight increasing trend in sea ice area (Figure 2). The anomalously low ice year of 1998 received considerable attention and the consensus was that low ice conditions within the northern route of the Northwest

Passage and the broader CAA were a result of an extended melt season together with an atmospheric circulation pattern that prevented sea ice from being transported southward through the CAA (Jeffers et al., 2001; Atkinson et al., 2006; Alt et al., 2006; Howell et al., 2010). After several consecutive low ice years, the ice area eventually recovered with an annual minimum of $116 \times 10^3$ km$^2$ in 2004 (Figure 2). Recovery in the years following 1998 has been attributed to seasonal FYI survival (i.e. FYI aging) and thicker, more resilient sea ice being imported into the northern route of the Northwest Passage from the M'Clure

Strait to the west and the Queen Elizabeth Islands to the north (Howell et al., 2023).





The spatial distributions of sea ice during the annual minimum for selected "low ice" years (here meaning: anomalously low for their decade) and the September 1991-2020 mean within the northern route of the Northwest Passage are shown in Figure 3. In the 1991-2020 mean, ice concentration typically remains high within the northwestern section of the Northwest Passage with lighter conditions found in the east (Figure 3j). In 1962, 1971, 1983, 1998, and 2007, there was some ice congestion the middle of Parry Channel and/or at the entrance to M'Clure Strait, despite overall ice concentrations being low along the passage (Figure 3a-e). These regions are known choke points within the Northwest Passage where MYI tends to accumulate despite low ice concentrations elsewhere or where MYI returns to first if they become cleared of thinner ice earlier in the summer (Cook et al., 2024). As result, Melling (2002) was the first to point out that persistent low ice conditions in the Northwest Passage would be difficult to achieve given that MYI gradually migrates southward through the CAA. Indeed, the amount of ice transiting southward from higher latitudes into the northern route of the Northwest Passage has reduced in magnitude since 2007 but there is no evidence of it stopping (Howell et al., 2023). However, for the record low year of 2024, as well as the low ice years of 2011 and 2023, low ice concentration was apparent across the entire route from east to west (Figure 3f; Figure 3h; Figure 3i). In these years, MYI transiting southward from higher latitudes was evidently insufficient to plug up the passage for the entire summer.

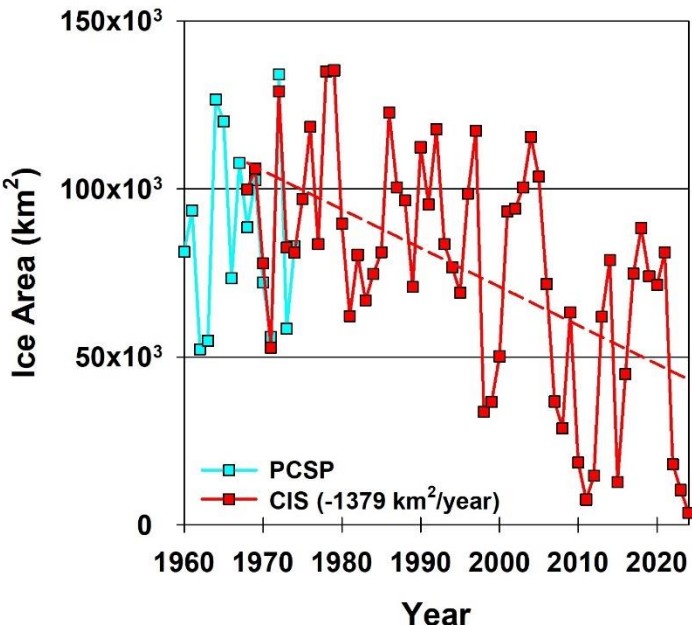

**Figure 2:** Time series of the week of minimum sea ice area in the northern route of the Northwest Passage from 1960-2024.





**Figure 3:** Spatial distribution of total ice concentration for the week of minimum sea ice area in the northern route of the Northwest Passage for a) 1962, b) 1971, c) 1983, d) 1998, e) 2007, f) 2011, g) 2015, h) 2023, i) 2024, and j) 1991-2020 mean.






## 3.2 Weekly evolution of sea ice in 2024 and other light ice years

The weekly time series of sea ice area in the northern route of the Northwest Passage for the low ice years of a) 2024, 2023, and 2011 b) 2007 and 1998, and c) 1983, 1971, and 1962 together with the 1991-2020 mean is shown in Figure 4. For the 1991-2020 mean, sea ice area during the pre-melt period (winter) was ~172x10$^3$ km$^2$ (i.e. covering the entire region) which decreased gradually during the spring and summer months of May to September to ~73x10$^3$ km$^2$ and then increased again in mid-September with the onset of freeze-up (Figure 4).


In 2024, sea ice area was already lower than the long-term mean during early summer, then rapidly declined until the end of August before gradually declining until the end of September, reaching the lowest observed sea ice area on record at 4x10$^3$ km$^2$ (Figure 4a). Particularly striking about 2024 was that freeze-up occurred considerably later (mid-October) when compared to other low ice years and the 1991-2020 mean (Figure 4a). The previous record low year of 2011 declined earlier faster than

2024 and ice area in 2011 was lower than 2024 when the minimum of 8x10$^3$ km$^2$ was reached in mid-September (Figure 4a). However, fall freeze-up started in mid-September in 2011 (Figure 4a). Interestingly, the weekly evolution of 2023 tracked lower than both 2011 and 2024 for a 2-week period in August but then remained relatively steady at 10x10$^3$ km$^2$ until freeze-up in early October (Figure 4a). The decline in 2024 (as well as in 2023 and 2011) was more substantial than in both 2007 and 1998 which reached minimum ice areas of 38x10$^3$ km$^2$ and 34x10$^3$ km$^2$, respectively (Figure 4a-b). Compared to the light ice

years in the early part of the CIS record (1962, 1971, and 1983), 2024 season was remarkable in the respect that summer minimum ice area was >50x10$^3$ km$^2$ lower, equivalent to ~30% of the area of the entire passage (Figure a; Figure 4c).

MYI area in the northern route of the Northwest Passage made up ~36% of the total ice area at the start of the melt season for the 1991-2020 mean and remained relatively stable throughout at ~52x10$^3$ km$^2$ with a slight increase at the end of September

and start of October (Figure 5a). The latter increase has been associated with ice import from higher latitudes together with seasonal FYI surviving the melt season and being promoted to MYI (Howell et al., 2023). Interestingly, this end-of-summer uptick in MYI did not follow the climatology in any low-ice year after 1998 but did in the earlier low-ice years of 1971 and 1983 (Figure 5). In 2024, MYI only made up ~10% of total sea ice area at the start of the melt season. During the 2024 melt season a slight increase in MYI occurred in early August but it was not sustained and MYI reached almost zero at the end of

the melt season (Figure 5a). For the previous record low year of 2011, MYI area at the start of the melt season was larger than in 2024 at 16% but it steadily declined to 2x10$^3$ km$^2$ with some replenishment during the early fall freeze-up season (Figure 5a). 2023 started the melt season with only 6% MYI showed but an overall increase occurred during the summer and fall (Figure 5a). Compared to the low ice years since 2007, in earlier years a key characteristic was that more MYI was present at the start of the melt season that did not decrease considerably during the melt season (Figure 5b-c). The exception was 1998

which had the largest amount of MYI present at the start of the melt season at about 50% that declined considerably but still remained high enough until mid-October to prevent the passage from becoming navigable (Figure 5b).




Sea ice area at the end of winter in 2024 was not significantly different to other years between 1991 and 2023, and the fraction of this ice that was MYI was not the lowest on record. However, the rapid rate of sea ice loss between July and September, the

continued loss until mid-October, and near-zero MYI replenishment in summer and fall, made the 2024 season remarkable with respect to other years. We now look at the driving factors pushing 2024 beyond the record low of 2011, and how the driving thermodynamic and dynamic processes in 2024 compared to other light ice years.

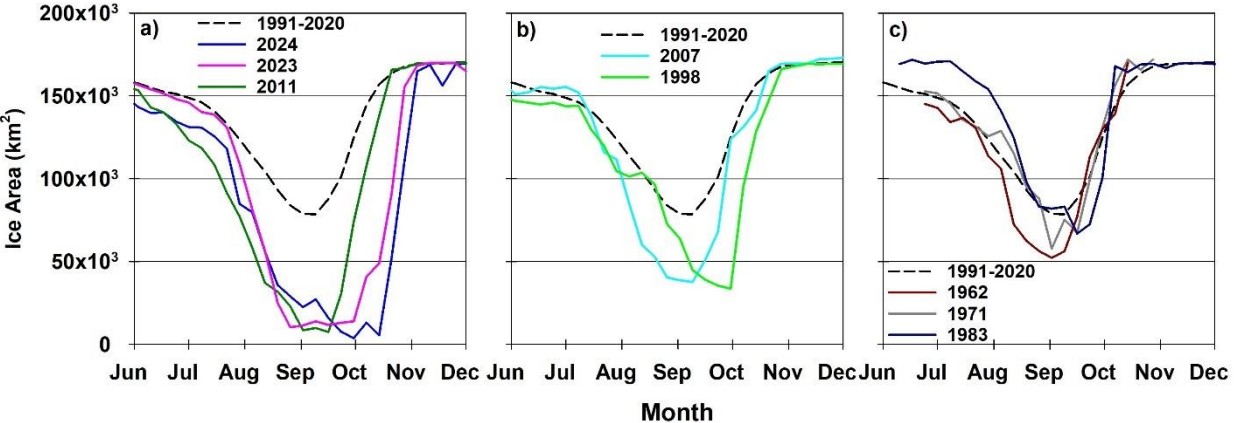

**Figure 4:** Weekly evolution of sea ice area in the northern route of the Northwest Passage for a) 2024, 2023, and 2011, b) 2007 and 1998, and c) 1962, 1971, 1983. The dashed line represents the 1991-2020 mean weekly evolution of sea ice area.

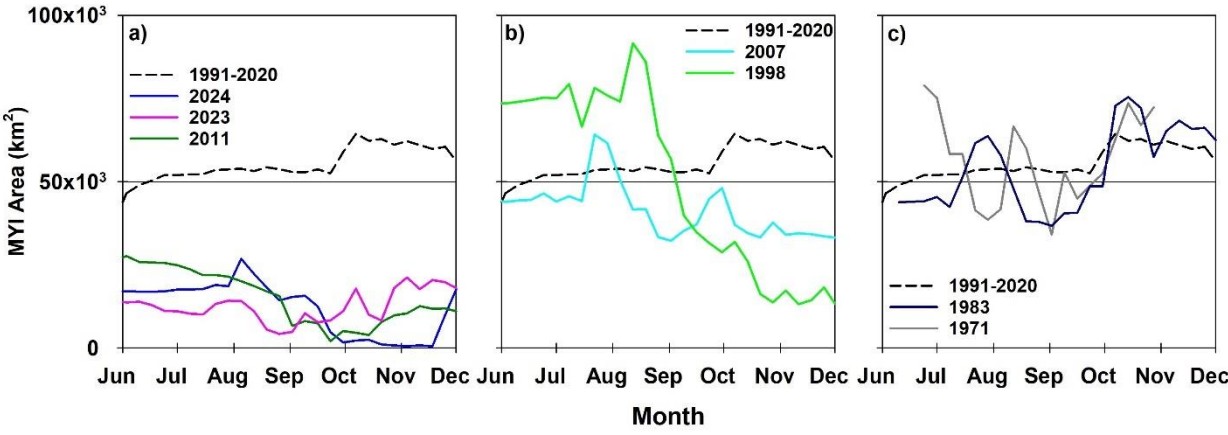

**Figure 5:** Weekly evolution of multi-year ice area in the northern route of the Northwest Passage for a) 2024, 2023, and 2011, b) 2007 and
1998, and c) 1971 and 1983. The dashed line represents the 1991-2020 mean weekly evolution of multi-year ice area.





## 3.3 Thermodynamic and dynamic forcing of summer sea ice area loss

It has been previously demonstrated that low ice years within the northern route of the Northwest Passage are driven by rapid melt associated with anomalously warm air temperatures together with an atmospheric circulation pattern that prevents sea ice from higher latitude regions from being imported into the region (Howell et al., 2009; Howell et al., 2013). The amount of MYI at the start of the melt season is also important as it is thicker and takes longer to melt through (Howell et al., 2009). Indeed, since 2007, low ice years in the northern route of the Northwest Passage have been a more common occurrence as a result of less MYI (Howell et al., 2023) but 2024 was remarkable considering the ice area almost reached zero at the end of the melt season. We suggest the 2024 record low in the northern route of the Northwest Passage was the result of the perfect sequence of forcing processes.

### 3.3.1. Thermodynamic forcing during light sea ice years

The spatial distribution of July to September (JAS) mean sea level pressure (SLP) anomalies over the Arctic is shown in Figure 6. 2024 was characterized by strong negative SLP anomalies over the western CAA (i.e. Banks Island and Victoria Island) that facilitated the flow of warm southerly air directly into the middle of the CAA (Figure 6). Accordingly, warmer air temperature anomalies were present across the southern regions of the CAA, including the northern route of the Northwest Passage in 2024 (Figure 7). This contrasts previous light ice years that are typically characterized by high SLP anomalies over Greenland (i.e. 1962, 1998, 2015, and 2023) that also facilitate the transport of warm southerly air into the CAA (Atkinson et al., 2006; Howell et al, 2010; Howell et al., 2013) (Figure 6; Figure 7). Indeed, warmer air temperatures were more widespread across the Arctic in 2023 however, the concentrated low SLP anomalies during JAS over the southwestern CAA in 2024 more effectively at transported warm air into the southern CAA compared to 2023 (Figure 8).

The time series of mean JAS air temperature anomalies over the southern CAA (i.e. regions in and to the south of Parry Channel) indicates that 2024 did in fact experience the highest temperature anomaly of 2.1℃, eclipsing previous years of 2023 (+1.8 ℃), 2011 (+1.5 ℃), and 1998 (+1.3℃) (Figure 9). In almost all the low ice years, there was a positive summer air temperature anomaly over the CAA, although it was slightly negative in the earliest years of 1962 and 1971, indicating that temperatures are not solely responsible for low ice years in the northern route of the Northwest Passage. Over the period of 1982-2024, air temperature in the southern CAA regions has increased by 0.45 ℃ decade$^{-1}$, increasing the likelihood of anomalously warm summers driving low ice years. Looking at the time series of mean JAS albedo anomalies indicates that 2024 ranked 2$^{nd}$ lowest at -0.16 with 2011 being the lowest at -0.15 and an overall decreasing trend of 0.02 decade$^{-1}$ from 1982-2024 (Figure 9). The statistically significant detrended correlation between the minimum sea ice area in northern route of the Northwest Passage and air temperature is -0.77 and -0.82 for albedo. Given all three of these variables have also experienced significant negative trends over the period of 1982-2024 suggests that minimum sea ice area in the northern route of the Northwest Passage is strongly related to warmer temperatures and lower albedo, over decadal timescales and



interannually. The record low year of 2024 was associated with the highest JAS anomaly together with the second lowest albedo anomaly that together helped enhance the sea ice albedo-feedback that facilitated the rapid melt in July and August (Figure 4a).

Although the rate of sea ice decline in 2024 slowed towards the end of August, it continued to decline well into October which
is indicative of an extended melt season (i.e. earlier melt onset and/or later freeze-onset) (Figure 4a). An earlier melt onset results in more energy absorbed into the sea ice during the melt season and helps enhance melt (Perovich et al., 2007). The timing of melt onset was not particularly early in 2024 when compared to the longer period of 2015-2024 (Figure 10) however, freeze-up in 2024 occurred almost 1-month later than the 1991-2020 mean (Figure 4). In fact, October air temperatures over the southern CAA experienced positive anomalies of upwards of 12℃ (Figure 11).

From the perspective of thermodynamic processes, 2024 was remarkable including a rapid melt period *and* an extended melt season pushing ice area to further extreme lows than previously observed. The coupling of a rapid melt event together with an extended melt season were two key thermodynamic factors responsible for 2024 reaching near zero ice area in the northern route of Northwest Passage.



**Figure 6:** Spatial distribution of 3-month mean July, August, and September sea level pressure (SLP) anomalies for a) 1962, b) 1971, c) 1983, d) 1998, e) 2007, f) 2011, g) 2015, h) 2023, and i) 2024. Anomalies are calculated with respect to the 1991-2020 July-August-September average.






**Figure 7:** Spatial distribution of mean July, August, and September 2 m air temperature anomalies for a) 1962, b) 1971, c) 1983, d) 1998, e) 2007, f) 2011, g) 2015, h) 2023, and i) 2024. Anomalies are calculated with respect to the 1991-2020 July-August-September average.





**Figure 8:** Spatial distribution of mean July, August, and September vertically integrated heat flux anomalies for a) 1962, b) 1971, c) 1983, d) 1998, e) 2007, f) 2011, g) 2015, h) 2023, and i) 2024. Anomalies are calculated with respect to the 1991-2020 July-August-September average. The magnitude is calculated as the vector magnitude of the northward and eastward anomalies.



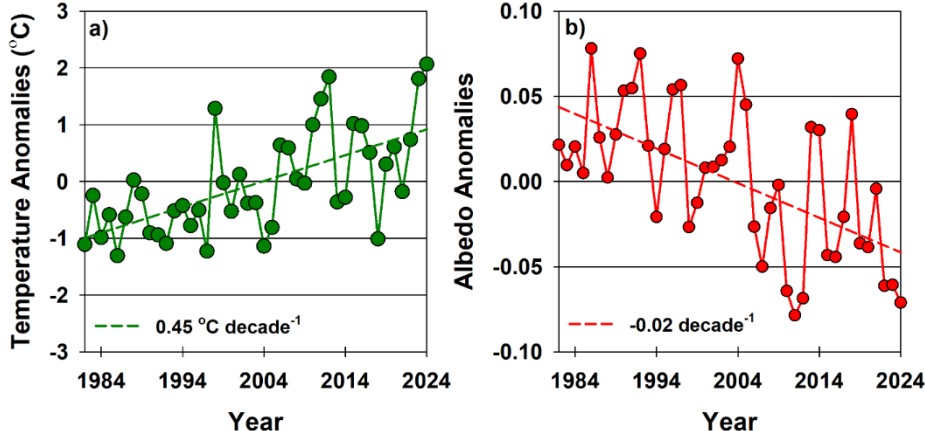

**Figure 9:** Time series of a) ERA5 2m air temperature anomalies and b) APP-x albedo anomalies from 1982-2024 in the southern region of the Canadian Arctic Archipelago.

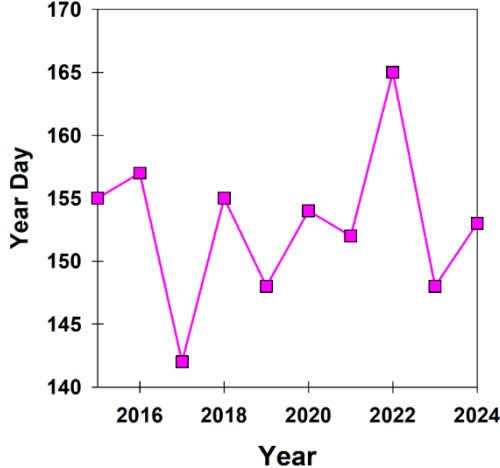


**Figure 10:** Time series of the timing of melt onset in the northern route of the Northwest Passage from 2015-2024.





**Figure 11:** Spatial distribution of mean October 2 m air temperature anomalies for a) 1962, b) 1971, c) 1983, d) 1998, e) 2007, f) 2011, g) 2015, h) 2023, and i) 2024. Anomalies are calculated with respect to the 1991-2020 July-August-September average.






### 3.3.2 Dynamic forcing during light ice years

Ice dynamics can encourage light ice conditions in the northern route of the Northwest Passage when ice import is low from adjacent regions (i.e. M'Clure Strait or the Queen Elizabeth Islands). Conversely, ice dynamics can prevent light ice conditions by facilitating ice import from these adjacent regions. The time series of the total May to September sea ice area flux for the M'Clure Strait and the Queen Elizabeth Islands from 2016 to 2024 is shown in Figure 12. The weekly time series of MYI presented in Figure 5 can also provide insight into ice import/export dynamic processes within the northern route of the

Northwest Passage as increases in MYI during the melt season or fall are typically associated with import from either the M'Clure Strait or the Queen Elizabeth Islands or both. An average $10\times10^3$ km$^2$ sea ice was imported into the northern route of the Northwest Passage from the M'Clure Strait and $14\times10^3$ km$^2$ from the Queen Elizabeth Islands for a total of $24\times10^3$ km$^2$ over the period 2016-2024. In 2024, only $2\times10^3$ km$^2$ of ice was imported from the M'Clure Strait and $9\times10^3$ km$^2$ from Queen Elizabeth Islands for a total of $11\times10^3$ km$^2$ (Figure 12), >50% below the average. The majority of this 2024 import occurred

in August as supported by the 2024 weekly time series of MYI (Figure 5a). 2023 experienced even less ice import from the M'Clure Strait and Queen Elizabeth Islands at $6\times10^3$ km$^2$ but record low ice conditions were averted because heat transport into the CAA was lower than in 2024 and the MYI area was higher (Figure 8; Figure 12). A lack of ice import alone is not sufficient to drive low ice area in the northern route of the Northwest Passage, rather it appears that – under current conditions within the CAA – that a combination of dynamic and thermodynamic factors must align. Although ice import during 2024 was

not the lowest over the period from 2016-2024 it was below average and that combined with the strong thermodynamic forcing (Figure 8) quickly ablated any imported ice leading to record low ice area conditions.

Comparing the ice dynamics of 2024 to the previous record low of 2011 indicates a similar process of limited import of ice during the melt season however, in 2011 ice area began to increase in late-September appreciably whereas in 2024 no increases

were observed until December (Figure 5). Sea ice motion is typically parallel to the SLP isobars (Thorndike and Colony, 1982) and Figure 6 indicates that although the SLP anomalies over the CAA are considerably different in 2024 compared to 2011, both acted to reduce the import of sea ice into the northern route of the Northwest Passage (Figure 6). In fact, compared to other low ice years in the northern route of the Northwest Passage only JAS SLP in 2007 was conducive to ice area import. Specifically, the high SLP anomalies over the Beaufort Sea in 2007 was found to drive sea ice into the northern Queen Elizabeth

Islands and subsequently southward into the northern route of the Northwest Passage (Howell et al., 2009).





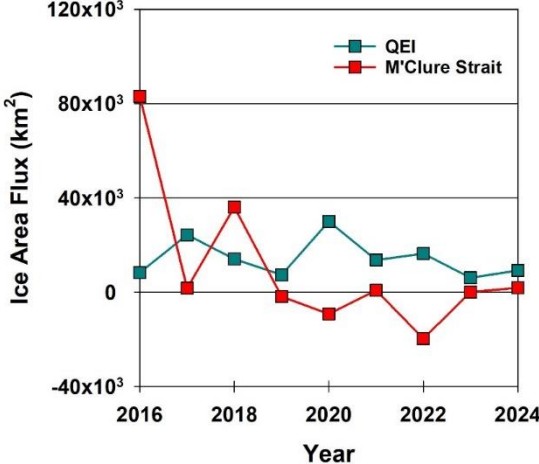

**Figure 12:** Time series the total May to September sea ice area flux at the M'Clure Strait and the Queen Elizabeth Island (QEI) southward gates from 2016 to 2024. Positive values indicate import and negative values indicate export.

### 3.4. Preconditioning

Another important characteristic that facilitates the clearing of the northern route of the Northwest Passage is the amount of MYI within the region because it is thicker and more resilient to melt (Howell et al., 2009). The resilience of MYI is best exemplified in 1998 when the northern route of the Northwest Passage contained 50% MYI at the start of the melt season that reduced the rate of ice loss (Figure 5b) despite anomalously high air temperatures during the summer months (Figure 8). MYI in both the southern and northern routes of the Northwest Passage have decreased by 33% from 2007 to 2020 relative to the 1991–2020 climatology (Howell et al., 2023). In 2024, the northern route of the Northwest Passage only contained about 10% MYI (compared to ~30% on average since 1991) with the remaining 90% being made up of seasonal FYI at the start of the melt season and, as a result, 2024 was certainly preconditioned to experience rapid ice area loss as and enhanced total melt from an extended melt season.

The thickness of the ice within the northern route of the Northwest Passage is also important because thinner ice will melt more rapidly than thicker ice under the same positive thermodynamic forcing. Pre-melt FYI thickness within the northern route of the Northwest Passage has likely decreased in recent years as Howell et al. (2016) found that maximum landfast FYI thickness at selected sites within the CAA decreased by ~25 cm since the late 1950s. MYI in the Arctic Ocean is also unlikely to be as thick now as it was historically, because it is generally younger (2 or 3 years of aging, rather than 2-6+ years historically) and has experienced significant year-on-year melting (e.g. Kwok, 2018; Kacimi and Kwok, 2022, Glissenaar et al., 2023; Sumata et al., 2023; Krumpen et al., 2025). Ice thickness observations are limited within the northern route of the Northwest Passage except for sporadic surveys conducted in 2011 and 2015 which do indicate the presence of thick ice (e.g. Haas and Howell, 2015) however, it is impossible to quantify how these years compare over a long-term record. We can




reasonably assume that the MYI present in the northern route of the Northwest Passage in recent years is thinner and less resilient, considering a wide-spread reduction in pan-Arctic sea ice thickness over recent decades.

## 4. Conclusions

At the end of the 2024 melt season, there was virtually no sea ice present along the northern route of the Northwest Passage. 2024 had the lowest observable ice area recorded since region-wide records began in 1960. This extreme event was driven by

the coupling of (i) rapid July-August melt, (ii) reduced ice import from higher latitude regions, and (iii) a 1-month extended melt season, together with (iv) a now-common, lower resilience FYI-dominated sea ice cover at the start of the melt season. Specifically, strong negative SLP anomalies over the western CAA and Beaufort Sea facilitated the flow of warm southerly air directly into the middle of CAA resulting in a record summer air temperature anomaly of +2.1°C that drove rapid melt. The same prevailing southerly winds only facilitated the net import of $11\times10^3$ km$^2$ of ice from higher latitude regions, 50% less

than the 2016-2024 mean. This was followed by an extended melt season where freeze-up was delayed by 1-month as a result of unprecedented +12°C air temperature anomalies in October.

The individual processes of rapid melt, an extended melt season, limited southward import and a less resilient spring ice cover have all occurred before and are unique characteristics of previous low ice area years. The difference with 2024 was that all

four occurred together whereas in previous years a maximum of three occurred together. For instance, 2011 experienced rapid melt, limited ice import together with less resilient spring ice but did not experience an extended melt season, and ice area began to increase again in late-September shortly after the annual minimum. 1998 experienced limited ice import and a long melt season, but the ice cover in the passage, composed of 50% MYI in spring, was more resilient during the melt season. 2007 experienced rapid melt together with less resilient ice but ice import during the melt season prevented record low

conditions.

In a warmer Arctic with a permanently lower resilience ice cover, sea ice within the northern route of the Northwest Passage is more susceptible to anomalous thermodynamic and dynamic forcing events that facilitate low ice years. However, in order for the northern route of the Northwest Passage to clear completely, several thermodynamic and dynamic processes need to

occur in sequence. Even low-ice years that do not lead to a completely clear the northern route of the Northwest Passage still require certain driving thermodynamic and dynamic processes to occur. Although some of these conditions are relatively predictable such as a low resilience ice cover or a relatively warmer summer air temperatures, others such an extended melt season and ice dynamics are not (Howell et al., 2020). Therefore, predicting ice conditions in the northern route of the Northwest Passage at leads times of several months remains difficult. Ice conditions along the northern route of the Northwest

Passage will likely continue to remain highly challenging considering ongoing variable during the transition to a summertime sea ice-free conditions which will continue to present significant risks to both reliable and safe navigation.



*Acknowledgments.* D. Babb is supported by the Canada Excellence Research Chair in Arctic Ice, Freshwater Marine Coupling and Climate Change held by D. Dahl-Jensen at the University of Manitoba.

*Data Availability.* A link to data access for the monthly sea ice area flux and melt onset timing via Environment and Climate Change Canada's Open Data Catalogue will be provided upon paper acceptance. PCSP ice charts are not yet publicly available but can be provided by contacting the SELH via email.

*Author contributions.* SELH wrote the manuscript with input from AC, DGB, JCL, JD, MM, and MB. SELH, AC, and MB
preformed the analysis. MM provided the melt onset data.

*Competing interests.* SEL is a member of The Cryosphere Editorial Board.

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
