# Peer review of "Near sea ice-free conditions in the northern route of the Northwest Passage at the end of the 2024 melt season"

_EGUsphere, 2025_

## Author Response (AR1)

Reviewer #1
The manuscript investigates conditions that led to record-low sea ice at the end of melt season in 2024. I do not have extensive background knowledge on the topic, but the paper is well-written, figures are generally of good quality, and the analysis is sound. In my opinion the manuscript can be accepted for final publication.

Howell et al.
Thank you. We enjoyed putting it together.

Technical comments:
Abstract, line 13, as well as line 63: It would be useful to give some reference value for sea ice area to compare with.

Howell et al.
Revised to:
Sea ice area in the northern route of the Northwest Passage on September 30, 2024 fell to a minimum of $4 \times 10^3$ km$^2$ or ~3% of its total area, the lowest ice area observed since 1960

Reviewer #1
Figure 1: A spatial scale bar would be useful. The scale bar for sea ice concentration is continuous, but the data in the map appears discrete.

Howell et al.
We added a scale bar. During the melt season there are a lot of partial ice concentration values which does not make for a consistent discrete legend year to year. The data is stretched for consistency (0 to 100%) for ease of display, especially for panel Figures.

Reviewer #1
Line 81: "complied" -> "compiled"

Howell et al.
Changed.

Reviewer #1
Line 93: "Sentintel" -> "Sentinel"

Howell et al.
Changed.

Reviewer #1
Figure 4: Gaps between x tick labels could be adjusted.

Howell et al.
They are spaced accordingly and easy to distinguish in our opinion.

Reviewer #2

This is a review of Near sea ice-free conditions in the northern route of the Northwest
The paper ice conditions and near sea-ice free conditions with respect to shipping prospects.
the paper is well elaborated and written. Thank you – it was great to read.

Howell et al.

Thank you. We enjoyed putting it together.

Reviewer #2

A few minor comments to be taken into account are found below.

- Please account for some spelling mistakes and some missing ",".

Howell et al.

We have tried to find all the type-o's.

Reviewer #2

- In Figure 1 a number of routes are shown. It is known that bathymetric information in the Arctic region are limited, but it would be good to indicate whether certain routes are affected by draft restrictions.

Howell et al.

Draft restrictions are a function of ship type. To that end, we did specify in the Introduction "shorter deepwater northern route directly through the Parry Channel from Baffin Bay to the Beaufort Sea and a shallow water southern route that runs south of Victoria Island." For more clarity, we have inserted "draft" before deepwater and shallow water.

Reviewer #2

- You might consider moving Figure 3 to the appendix, as Figure 2 contains all relevant information and Figure 3 distracts somewhat the focus. Nevertheless, it is useful to be part of the work.

Howell et al.

Indeed, but Figure 3 contains the spatial distribution of minimum ice conditions for the various light ice years and we feel this is very important to showcase.

Reviewer #2

- One item not addressed are fragmented ice floes or multi-year ice pieces floating close to the ice edge. With the methods applied this information are generally difficult to obtain, but the authors might at least address it in the discussion. The background is that random floating ice pieces as well as the marginal ice zone are considered hazardous for ships especially, when not reinforced for ice contact or with limited reinforcement. The ice extend is a parameter of significance, but not the only one. As mentioned, this might require different methods, but should be mentioned.

Howell et al.

We agree but any floe of multi-year ice is hazardous to a ship. We feel it does not really warrant discussion because our paper focuses the processing driving light ice years but we make this point clearer in the introduction as follows: "This is because there are regions along the northern route of the Northwest Passage that contain high concentrations of MYI which is the most

hazardous to ships during the summer months and a direct path east to west is typically not possible (Cook et al., 2024)."

Community Comment by Wang Zihan
This manuscript is already very mature in its structure, datasets, and analysis. The authors make good use of multiple sources and provide a thorough and convincing examination of the 2024 extreme low-ice conditions in the Northwest Passage. The findings are both timely and valuable. My comments below are offered in a constructive spirit, as possible ways to further strengthen the paper.

Howell et al.
Thanks for the comments. We have incorporated them into a revised version.

Major Comment
The manuscript already provides a very solid analysis of atmospheric and dynamic factors. To gain an even deeper understanding of the extreme conditions in 2024, it could be valuable to also consider the oceanic environment during the melt season. Variables such as sea surface temperature or upper-ocean heat content, in the same way as air temperature and sea level pressure are analyzed, may provide additional insight into why the freeze-up was so strongly delayed. Datasets such as ORAS5 (since 1958, though its 0.25° resolution may be coarse for the CAA) and GLORYS (since 1993, 0.083°) could be useful for this purpose. Adding this dimension would further enrich the physical interpretation of the 2024 extreme event.

Howell et al.
Our intent is to focus on the drivers of ice loss in the northern route of the Northwest Passage and SST is more of an indicator or an association. Further, it is widely known that high SSTs are often associated with delayed freeze-up as a result of a strong ice-albedo feedback. Freeze-up in 2024 was delayed because of an enhanced sea ice albedo-feedback and warmer October temperatures. Indeed, SST should be mentioned but we do not feel additional panels are necessary. We added SST context based on the recent analysis from Timmermans and Labe (2025) from the recent Arctic Report Card as follows: "Positive sea surface temperature (SST) anomalies within the northern route of the Northwest Passage were also present during September (Timmermans and Labe, 2025) providing additional support for the enhanced sea ice albedo-feedback during the summer months."

Reference:
Timmermans, M.-L. and Labe, Z.: Sea surface temperatures. [in "State of the Climate in 2024"]. Bull. Amer. Meteor. Soc., 106 (8), S320–S3322, https://doi.org/10.1175/BAMS-D-25-0104.1, 2025

**Minor Comments**
   1. L28: At this location, please also write Northwest Passage in full rather than the abbreviation NWP, for consistency.
Done

2. L62: Remove the duplicate "in in".

Removed.

3. L76: The phrase "is available at and the data is …" is redundant; simplify to "is available at: …".

Changed.

4. L64: The sentence "the lowest ever observed area of since 1960" should remove "of".

Removed.

5. L174: The phrase "2011 declined earlier faster than 2024 …" could be revised to "declined earlier and faster than in 2024".

Revised.

6. Throughout the text, "2m air temperature", "2 m air temperature", and "2-metre air temperature" appear. Please unify to one consistent style.

Unified.